# Dysbacteriosis-Derived Lipopolysaccharide Causes Embryonic Osteopenia through Retinoic-Acid-Regulated DLX5 Expression

**DOI:** 10.3390/ijms21072518

**Published:** 2020-04-04

**Authors:** Lingsen You, Liwei Zhu, Pei-zhi Li, Guang Wang, Hongmei Cai, Jinhuan Song, Denglu Long, Zachary Berman, Li Lin, Xin Cheng, Xuesong Yang

**Affiliations:** 1Division of Histology and Embryology, Joint Laboratory for Embryonic Development & Prenatal Medicine, Medical College, Jinan University, Guangzhou 510632, China; Lingsenyou@gmail.com (L.Y.); zliwei12138@126.com (L.Z.); li_peizhi@gibh.ac.cn (P.-z.L.); wangguang7453@126.com (G.W.); caihongmei14020@gmail.com (H.C.); sjh1207359577@gmail.com (J.S.); longdenglu123@gmail.com (D.L.); linli369963@163.com (L.L.); 2Department of Radiology, University of California San Diego, San Diego, CA 92103, USA; ztberman@gmail.com; 3Key Laboratory for Regenerative Medicine of the Ministry of Education, Jinan University, Guangzhou 510632, China

**Keywords:** microbiota dysbiosis, LPS, lipopolysaccharide, embryos, bone formation, osteopenia, *Dlx5*, RA, retinoic acid

## Abstract

Growing evidence suggests an adverse impact of gut microbiota dysbiosis on human health. However, it remains unclear whether embryonic osteogenesis is affected by maternal gut dysbacteriosis. In this study, we observed that elevated lipopolysaccharide (LPS) levels led to skeletal developmental retardation in an established mouse model of gut microbiota dysbiosis. Using chick embryos exposed to dysbacteriosis-derived LPS, we found restriction in the development of long bones as demonstrated by Alcian blue and alizarin red staining. Micro-CT and histological analysis exhibited decreased trabecular volume, bone mineral density, and collagen production, as well as suppressed osteoblastic gene expression (*Ocn*, *Runx2*, *Osx*, and *Dlx5*) in chick embryonic phalanges following LPS treatment. Atomic force microscopy manifested decreased roughness of MC3T3-E1 cells and poorly developed matrix vesicles (MVs) in presence of LPS. The expression of the aforementioned osteoblastic genes was suppressed in MC3T3-E1 cells as well. High-throughput RNA sequencing indicated that retinoic acid (RA) may play an important role in LPS-induced osteopenia. The addition of RA suppressed *Dlx5* expression in MC3T3-E1 cells, as was also seen when exposed to LPS. Quantitative PCR, Western blot, and immunofluorescent staining showed that retinoic acid receptor α (RARα) was upregulated by LPS or RA treatment, while the expression of DLX5 was downregulated. CYP1B1 expression was increased by LPS treatment in MC3T3-E1 cells, which might be attributed to the increased inflammatory factors and subsequently activated NF-κB signaling. Eventually, blocking RA signals with AGN193109 successfully restored LPS-inhibited osteoblastic gene expression. Taken together, our data reveals that maternal gut microbiota dysbiosis can interfere with bone ossification, in which *Dlx5* expression regulated by RA signaling plays an important role.

## 1. Introduction

There are 300–500 different strains of bacteria colonized in each human intestinal tract [1]. Human health is closely linked to the commensal gut microbiota which establishes a complicated interaction between the bacterial communities and human host [2]. These resident bacteria help prevent invasion of alien microbes, provide nutrition to intestinal epithelium, and promote immune/neuroendocrinal modulation under normal physiological conditions [3,4], suggesting that the gut microbiota exerts a pivotal function on regulating host processes [5]. An increasing number of studies have shown that the microbial community influences host developmental processes such as organ development and morphogenesis, as well as normal development of the fetus during pregnancy [4]. Homeostasis between host and gut microbiota is maintained by constant intense crosstalk. However, dysbiosis of gut microbiota is not rare and is worsened with diet, obesity, excessive stress or inflammation, insulin resistance, metabolic syndromes, and antibiotic misuse [6,7].

In the setting of dysbiosis, the microbiota and their metabolites could lead to the development of pathogens. With pathogenic bacterial overgrowth, there is local intestinal inflammation which may lead to many other systemic diseases via the complex interplay between human host and bacteria [8]. Lipopolysaccharide (LPS), the major component in the outer membrane of Gram-negative bacteria, is found to be increased in the blood with gut microbiota dysbiosis [9,10]. When the concentration of LPS is increased to two to three times that of normal values, it reaches the threshold of metabolic endotoxemia, which may interfere with several aspects of normal host physiological function [11,12]. Previous studies have proven that an elevated maternal LPS can lead to LPS reaching fetuses by passing through the placental barrier [13]. Once through this barrier, the microbiota-dysbiosis-derived LPS could negatively affect fetal development and cause intrauterine fetal demise in severe cases [14].

Toll-like receptor 4 (TLR4), the innate immune receptor of LPS, interacts with LPS to activate NF-κB signaling to induce subsequent biological effects [15]. There have been many reports about the deleterious effects of maternally derived LPS, such as leading to embryonic death [16,17], impairing spermatogenesis in mice offspring [18,19], negatively impacting blastocyst implantation, and causing fetal growth retardation or malformation [19,20,21]. However, the knowledge of the effect of maternally-derived LPS during gestation on embryonic osteogenesis is quite limited. This is important given that skeletal abnormalities are usually accompanied with intrauterine growth retardation [22]. The Developmental Origins of Health and Disease (DoHaD) hypothesis suggests that low birth weight can permanently influence or “program” long-term development and disease in later life. From this perspective, a more complete insight of the fetal development is of importance to understand the pathogenesis of adult disease [23,24].

Vertebrate skeletal development is accomplished by either endochondral or intramembranous ossification. Osteoblasts, originating from mesenchymal cells, are responsible for bone matrix formation, playing a central role during bone development [25]. Advances in skeletal genetics have identified key regulators of osteoblast differentiation and function, such as Runt-related transcription factor 2 (*Runx2*) and osterix (*Osx*) [26]. Recent observations have identified distal-less homeobox 5 (*Dlx5*) as an important transcriptional factor involved in determining osteoblast lineage differentiation as well as modulating *Runx2* and *Osx* gene expression. *Dlx5*, expressed from the very early stages of bone development, eventually disappears in fully differentiated osteocytes and acts at multiple stages of osteogenesis. Several studies have highlighted that *Dlx5* plays a vital role in the modulation of osteogenesis by controlling the expression of bone-related genes [26,27,28].

In this present study, mouse and chick embryos were employed to explore the influence of gut microbiota dysbiosis derived LPS exposure on osteogenesis. To address the corresponding molecular mechanisms, RNA sequencing techniques were used to detect changes in signaling pathways at the early stages, when *Dlx5* is beginning to be specifically expressed in the ectoderm of the presumptive limb-forming regions of the lateral plate, but not in the intervening non-limb-forming prospective flank [29]. Our results show that LPS exposure leads to changes in the retinoid metabolism pathway, which implies that retinoic acid (RA) might affect the function of *Dlx5*. Next, in vitro cell culture experiments demonstrated that the elevation of RA leads to a downregulated expression of *Dlx5*, which subsequently impaired bone formation.

## 2. Results

### 2.1. Retardation of Embryonic Skeleton Development Was Observed in Both Pregnant Mice with Lincomycin Hydrochloride (LH)-Induced Dysbacteriosis and Chick Embryos Treated by LPS

To investigate the effects of maternal dysbacteriosis on embryonic skeleton development, we created a dysbacteriosis mouse model through injection of LH (Figure 1A). LH treatment led to an increase of Gram-negative bacteria and reduction of Gram-positive bacteria. The imbalance of intestinal flora was associated with an elevated level of LPS in the serum, which suggested that the mouse model was successfully established (Figure 1B,C). Compared to control E14.5 mice embryos, embryonic sizes and body weights, as well as fore- and hind-limb lengths, were all significantly reduced in the LH-treated group (Figure 1D–H). Next, we mimicked the LH-induced mouse model with chick embryos by exposing to LPS (Figure 1I). Higher mortality, lower body weights, and elevated levels of serum LPS were seen in LPS-treated chick embryos when compared to controls (Figure 1J–L). Alcian blue–alizarin red double-staining demonstrated that LPS treatment led to smaller skeletons of E17 chick embryos and shorter long bones, such as the tibia, femur, humerus, ulna, radius, and phalanges (Figure 1M–T), suggesting that dysbacteriosis-induced LPS elevation could appreciably arrest embryonic osteogenesis.

To determine the effects of dysbacteriosis on ossification, we implemented micro-CT on E17 chick phalanges (Figure 2A,B). This showed that LPS exposure was associated with the reduction of relative bone volume (bone volume fraction, BV/TV), trabecular numbers (Tb.N), trabecular thickness (Tb.Th), bone mineral density (BMD), enhanced porosity, and trabecular separation (Tb.Sp) (Figure 2C–H). Hematoxylin and eosin (H&E), Picrosirius red, and von Kossa staining was performed on the longitudinal sections of phalanges to quantitatively measure relative bone area, collagen type I (COL1) area, and mineralized area. This demonstrated significant suppression of bone formation by LPS treatment (Figure 2I–N). Immunofluorescent staining found that LPS treatment dramatically inhibited COL1 expression (Figure 2O,P); similar results were obtained by quantitative PCR (qPCR), in which collagen type I alpha 1 (*Col1α1*) and alpha 2 (*Col1α2*) expression at mRNA level was suppressed in the presence of LPS (Figure 2Q,R). The ratio of *Col1a1*/*Col1a2* was also notably affected by LPS exposure (Figure 2S). These data imply that dysbacteriosis-induced LPS elevation may cause the decreasing bone mass formation during embryonic osteogenesis.

### 2.2. Downregulation of DLX5 Was Responsible for Aberrant Skeleton Development Induced by LPS Exposure in Chick Embryos

To explore the mechanism of aberrant skeleton development induced by dysbacteriosis-elevated LPS, we evaluated the expression of osteogenesis-related genes in E17 chick embryos exposed to LPS (Figure 3). Combined immunofluorescent staining and Western blot data showed that osteocalcin (OCN/BGLAP) expression was significantly depressed in presence of LPS (Figure 3A,B). qPCR confirmed that the expression of *Ocn* and also alkaline phosphatase (*Alpl*/*Tnap*), osteonectin (*On*/*Sparc*), and bone sialoprotein (*Ibsp*) were all downregulated in the LPS-treated group. Osteopontin (*Opn*/*Spp1*) was not depressed (Figure 3C). Immunofluorescent staining of RUNX2 and OSX demonstrated that their expressions in the E17 chick phalanges were dramatically suppressed by LPS treatment (Figure 3D,E). Analysis with the STRING (Protein–Protein Interaction Networks, Functional Enrichment Analysis) database demonstrates that DLX5 plays a vital role among the protein–protein interactions during osteoblastogenesis through its control of the expression of downstream genes such as OCN, ON, IBSP, RUNX2, and OSX (Figure 3G). Subsequent qPCR, immunofluorescent staining, and Western blot data confirmed that DLX5, RUN2, and OSX were downregulated in presence of LPS (Figure 3D–F,H,I).

To investigate the possible underlying reasons for suppressed osteogenic differentiation, we studied the morphology of osteoblasts following exposure to LPS in vitro (Figure 4). F-actin staining showed the morphological change of MC3T3-E1 cells following LPS exposure after 7 days (Figure 4A,B). Atomic force microscopy (AFM) data demonstrated that LPS treatment increased the ratio of cell length to width, while Rq (root-mean-square roughness) and Ra (average roughness) of the MC3T3-E1 cells were significantly decreased in presence of LPS compared to controls (Figure 4C–E). Under AFM, the heights of the released MVs were considerably reduced in the presence of LPS (Figure 4F,G). This was further confirmed by transmission electron microscopy (TEM) analysis on the diameter of the MVs, as well as the mRNA expression of MV-associated markers phosphoethanolamine/phosphocholine phosphatase 1 (*Phospho1*), tissue nonspecific alkaline phosphatase (*Tnap*), ectonucleotide pyrophosphatase/phosphodiesterase 1 (*Npp1*), *CD63*, and *CD9* (Figure 4H,I).

We next cultured MC3T3-E1 cells in osteogenic differentiation medium with the addition of LPS to evaluate the effects of LPS on osteoblastic function. The results of alizarin red staining showed that LPS significantly decelerated the mineralization of MC3T3-E1 micromass cultures, as well as monolayer cultures, when compared with controls (Figure 5A). Meanwhile, the expression of OCN was significantly suppressed at day 7 by LPS exposure (Figure 5B–D). qPCR data further showed that the expressions of *Ocn*, *On*, *Alpl*, *Ibsp*, *Col1*α*1*, and *Col1*α*2* were suppressed by LPS treatment while the ratio of *Col1α1*/*Col1α2* increased (Figure 5E,F). OSX, RUNX2, and DLX5 immunofluorescent staining as well as qPCR showed that LPS treatment inhibited these gene expressions at both mRNA and protein levels in MC3T3-E1 cells (Figure 5G–O). Taken together these in vitro data suggest that LPS exposure inhibited osteogenesis through its effect on the expressions of these vital osteogenic genes, as was also seen in vivo.

### 2.3. RA Metabolism Played an Important Role on LPS-Induced Abnormal DLX5 Expressions

To fully elucidate the mechanism of abnormal osteogenic gene expressions, we next focused on DLX5, the activator of the transcription factors regulating osteogenesis. DLX5 exerts function from the very early stages of bone development. We implemented RNA sequencing to screen for gene expression changes in E1.5 chick embryos exposed to LPS. GO annotations analysis was used to establish the affected signal pathway caused by LPS exposure, including those related to the biological processes, cellular components, and molecular functions (Figure 6A). The genes involved with development and metabolism proved to be closely related to dysbacteriosis. This was further enriched by KEGG analysis which revealed that retinoid metabolism underwent a significant change following LPS treatment (Figure 6B). This was subsequently verified by the results determining the RA level in both E17 chick phalanges and MC3T3-E1 cells exposed to LPS (Figure 6C,D). Subsequently, DLX5 expression was checked in MC3T3-E1 cells exposed to RA, showing a similar tendency as was seen with LPS treatment (Figure 6E,F).

To address how RA signaling regulates *Dlx5* expression, we determined the expressions of RA receptors (*RARs*) and retinoid X receptors (*RXRs*), the heterodimers mediating retinoid gene expression. qPCR data demonstrated high levels of *RARs* (α, β, γ) and *RXRs* (α, β, γ) in MC3T3-E1 cells when exposed to LPS (Figure 7A–F). Next, the most notably changed subtype, RARα, was double immunofluorescent stained with DLX5. This showed that the exposure to either LPS or RA could inhibit DLX5 expression and enhance RARα expression in MC3T3-E1 cells (Figure 7G,H). To illuminate the mechanism of how LPS induced the increase of RA, we checked the RA-metabolism-related enzymes’ gene expression using qPCR, immunofluorescent staining. and Western blot. This found that the expressions of one of RA synthetase-cytochrome P450 1B1 (*Cyp1b1*) were upregulated by LPS exposure, while the expression of another synthetase family, retinaldehyde dehydrogenases (*Raldhs*), did not change significantly. The cytosol-to-nuclear shuttling protein, cellular retinoic-acid-binding protein 2 (*Crabp2*), facilitating RA binding to its cognate receptor complex and transferring to the nucleus, was upregulated as well while another shuttling protein, fatty-acid-binding protein 5 (*Fabp5*) did not show significant change (Figure 7I–P). Furthermore, the gene expressions of RA catabolic enzymes (cytochrome P450 26 subtypes *Cyp26a1*, *Cyp26b1* and *Cyp26c1*) were increased (Appendix A).

Eventually, the addition of AGN, a potent pan-RAR antagonist, could efficiently reverse the LPS-induced downregulations of *Dlx5*, *Runx2*, *Osx*, *Ocn*, *Alpl*, *Ibsp*, and *Col1α1* in MC3T3-E1 cell cultures, as demonstrated by qPCR, immunofluorescent staining, or Western blot (Figure 8). These data further confirm that RA plays a crucial role on *Dlx5*-mediating osteoblastic differentiation driven by the elevated RA in presence of LPS.

### 2.4. LPS-Induced Inflammatory Cytokines Lead to RA Increases in Osteoblasts

We speculated that the elevation of CYP1B1 expression in presence of LPS might be associated with the activation of interleukin 6 (IL-6) and tumor necrosis factor alpha (TNF-α), as previously reported in other cell lines [30,31]. In MC3T3-E1 cells, LPS treatment increased nuclear translocation of P65 (Figure 9A–C). qPCR showed that LPS activated the TLR4/NF-κB signaling pathway by upregulating *Tlr4*, *CD14*, *Myd88*, *P65*, *IL-6*, and *TNF-α*, as well as downregulating *IκBα* (Figure 9D–I). The upregulation of IL-6 and TNF-α was also verified by ELISA assay (Figure 9J–K). These results suggest NF-κB signaling activation and the subsequent inflammatory cytokine upregulation lead to the RA increase in osteoblasts when exposed to LPS.

## 3. Discussion

Growing data show that intestinal dysbacteriosis is associated with many disorders commonly observed in clinical situations. In pregnant women, intestinal microbiota dysbiosis could cause metabolic endotoxemia and elevate blood LPS levels, which can pass through the placental barrier, negatively influencing the developing fetus [13,21,32]. LPS-induced developmental toxicity at early gestational stages has been previously characterized as well [33]. We have also reported that LPS exposure during gestation could restrict the conversion of chondrocytes from proliferating to hypertrophic in the growth plate [34]; however, it is still obscure whether embryonic osteogenesis is affected by the LPS exposure during gestation.

In this study, we successfully established a mouse model of gut microbiota dysbiosis, which could be reflected by the enhanced portion of Verrucomicrobiae and flora dysbiosis in mouse feces. In this dysbiosis model, we found that the growth of mouse embryonic limbs was retarded. Further experiments were performed on LPS-treated chick embryos to mimic gut microbiota dysbiosis because chick embryos can be cultured in vitro from stage HH0, which makes the manipulation of early stage embryos possible. Furthermore, the chick embryo works as a good model of skeletogenesis, since its skeletal size is larger than mice embryos of the same embryonic stage. Utilizing chick embryos avoids the complicated bidirectional interactions between mother and fetus seen in mammals and may help elucidate the potential detrimental effect of microbial dysbiosis on fetal development [35].

Gut microbiota dysbiosis induced mouse skeletal abnormalities were replicated in LPS-treated chick embryos (Figure 1). LPS treatment led to the significant reduction of trabecular number, trabecular thickness, bone mineral density, and collagen secretion (Figure 2). These findings indicate that developmental osteopenia is enhanced after LPS exposure during the fetal period. Together, the data suggested that gut microbiota dysbiosis affects embryonic osteogenesis, which may be principally due to the restriction of bone formation, given that we did not find noticeable bone absorption of chick phalanges in the presence of LPS (Appendix A).

Bone formation is mainly achieved by osteoblasts which produce the bone matrix and participate in mineral deposition. We showed that the proliferation of osteoblasts was not significantly changed after exposure to LPS, as demonstrated by immunofluorescent double staining of PCNA (proliferative cellular nuclear antigen) and OCN, the marker of osteoblasts (Appendix A). Therefore, we checked the expression of the functional markers of osteoblasts. First, LPS exposure remarkably inhibited the expressions of the genes regulating bone mineral maturation (*Ocn* and *On*) [36], as well as *Alpl*, a reliable indicator for developmental maturity of mineralized tissue cells [37] (Figure 3A–C). The expressions of *Runx2* and *Osx*, vital transcription factors regulating osteoblastic differentiation [38], were significantly depressed in the bone trabecula of phalanges exposed to LPS (Figure 3D–E). Growing attention in the literature has been directed at DLX5, which has been reported to control the expressions of numerous osteogenetic genes [26,38,39]. This may be the core factor affected in the restriction of bone formation induced by microbiota-dysbiosis-elevated LPS, as we demonstrated that the expression of DLX5 in the periosteum, bone trabecula, and bone marrow was suppressed dramatically by LPS treatment (Figure 3F).

To elucidate how osteoblastic differentiation was inhibited, we exposed MC3T3-E1 cells to LPS following a screening for optimal in vitro conditions (Appendix A). AFM demonstrated that LPS treatment suppressed the osteoblastic differentiation by decreasing Ra and Rq of MC3T3-E1 cells and withering the size of MVs (Figure 4). MVs are extracellular structures which are believed to be one of the sites of mineral nucleation that occurs in the organic matrix of the skeletal tissues [40]. The morphological properties of MVs are closely associated with osteoblast precursor differentiation. These results indicated that LPS inhibited the mineralization process enhanced by MVs. It was further confirmed that the mineralization capacity of MC3TC-E1 cells (Figure 5A) and osteoblastic gene expressions were also suppressed by LPS exposure (Figure 5B–E). As an upstream regulator, DLX5 expression was decreased in presence of LPS, along with inhibited expressions of RUNX2 and OSX, the bone-formation-specific transcription factors [41] (Figure 5G–O). This implies that abnormal expressions of DLX5 assume an indispensable role in the restriction of embryonic bone formation induced by microbiota dysbiosis.

To address how elevated LPS levels affect *Dlx5* expression, RNA sequencing of HH10 chick embryos exposed to LPS was performed with transcriptome-wide analysis of differential genes. HH10 chick embryos were utilized because *Dlx5* was previously reported to be expressed from the HH10 stage onwards [29]. An intimate correlation exists between microbiota and systemic metabolism [42]; therefore, we chose the metabolic- and developmental-relevant genes for further study by GO annotation analysis. Of the altered genes seen and combined with prior literature, the most likely candidate is the RA signaling pathway [43,44,45]. RA is a vitamin-A-derived, nonpeptidic, structurally simple lipophilic molecule, which plays a pleiotropic role during development. Increased levels of RA were seen in E17 chick phalanges and MC3T3-E1 cells exposed to LPS (Figure 6A–D). Furthermore, treating MC3T3-E1 cells with RA dramatically reduces DLX5 expression, implying that RA may negatively regulate DLX5 expression (Figure 6E–F).

Lastly, we sought to answer how elevated levels of LPS influence RA signaling. RA acts as ligand for nuclear RARs and RXRs, converting them from transcriptional repressors to activators, which in turn regulate gene expression [46,47]. We discovered that the expressions of all RARs, especially RARα, were significantly enhanced by LPS exposure, and DLX5 expression significantly decreased in the presence of LPS as well as RA (Figure 7A–H), indicating that RA signaling was able to regulate *Dlx5* expression either in a direct or indirect way. Our next concern was to explain how the RA level was elevated. The concentration of RA in tissues is dependent on the biosynthesis and degradation of retinoid metabolism, which is controlled by retinaldehyde dehydrogenases and specific cytochrome P450 enzymes (CYP1B1 and CYP26s) [46]. As previously noted, there was a change of cytochrome P450 enzyme levels after exposure to LPS, as determined with RNA sequence analysis. Our experiments demonstrated that the upregulation of CYP1B1, the key enzyme in RA biosynthesis, was mainly responsible for RA elevation in the presence of LPS. Based on previous studies in hepatocytes and a tumor cell line [30,31], proinflammatory cytokines IL-6 and TNF-α are involved in regulation of CYP1B1 expression. These inflammatory mediators are also mediated by LPS. With the elevated levels of LPS used in our model, we speculated that inflammatory cytokines might work to cause the upregulation of CYP1B1. This was confirmed by the activation of NF-κB signaling pathway in MC3T3-E1 cells exposed to LPS. IL-6 and TNF-α were significantly upregulated simultaneously (Figure 9). Eventually, the inhibition of RA signaling with AGN successfully restored the expressions of key osteogenic genes, including DLX5, RUNX2, OSX, and OCN, in MC3T3-E1 cells (Figure 8). This further verified the vital modulatory role of RA signaling on influencing DLX5 expression and bone ossification. We could not completely ascertain whether the regulatory role of RA was to activate the promoters of *Dlx5* directly. However, we are prone to believe it is, due to the ability to predict the binding sites of RAR/RXR dimers known as RAR elements (RAREs) in upstream flanking region of the exon of *Dlx5* as seen on NUBIScan [48] (Appendix A).

In summary, we proposed that gut microbiota dysbiosis led to the elevated level of LPS in maternal serum. The developing embryo/fetus would be exposed to the maternally derived LPS, which can easily pass through the placental barrier due to its lipophilic nature. LPS interfered with embryonic bone ossification through activating NF-κB signaling and stimulating the releases of IL-6 and TNF-α. These inflammatory cytokines caused the increase of CYP1B1 expression and then subsequently activated RA signaling, which probably inhibited the transcription of *Dlx5* directly (Figure 10), and eventually restricted embryonic bone ossification. There is no doubt that more precise experiments are required to explore the underlying molecular mechanisms in the future.

## 4. Materials and Methods

### 4.1. Ethic Statement

All procedures and protocols were approved by the Institutional Animal Care and Use Committee (IACUC) of Jinan University (IACUC-20181126-02, 26 November 2018) and were conducted in accordance with the ethical standards of the Helsinki Declaration.

### 4.2. Experimental Mouse and Chick Embryos

Mice: The C57BL-6 mice used in this study were obtained from the Laboratory Animal Center of Sun Yat-Sen University (Guangzhou, China). All mice were maintained under environmentally controlled conditions, 23 ± 2 °C, with relative humidity of 55% ± 10%, under a 12-h light/dark cycle, with free access to standard laboratory diet and water. Five-week-old female mice were used to induce gut microbiota dysbiosis by injecting saline (control, 0.9%, 20 mL/kg body weight) or lincomycin hydrochloride (LH, 0.3 g/mL, 24 mL/kg; Shanghai Macklin Biochemical Co., Ltd.) for 7 days consecutively. Two female mice were housed with one normal male mouse overnight in a cage. The day the vaginal plugs were first observed was designated as embryonic day 0.5 (E0.5); embryos were harvested on E14.5 (Figure 1A).

Chick embryos: Fertilized Leghorn eggs were obtained from the Avian Farm of the South China Agriculture University (Guangzhou, China), and incubated until reaching the required developmental stage (Hamburger and Hamilton, HH) in a humidified incubator. For LPS or saline (control) exposure, 200 µL of 10 µg/mL LPS (Sigma–Aldrich, St. Louis, MO, USA) or 0.71% sterile saline were carefully injected into a small hole made in the air chamber of the eggs every other day from E1.5 to E17. The surviving embryos were harvested for further analysis.

### 4.3. High-Throughput Sequencing Analysis

The intestinal flora DNA was extracted and purified from fecal samples of pregnant mice by the E.Z.N.A. soil DNA Kit (Omega Bio-tek, Norcross, GA, USA) [49]. DNA was diluted to 1 ng/μL using sterile water. The 16S ribosomal RNA (rRNA) gene was analyzed by Illumina Miseq (Novogene Technology Co., Ltd., Beijing, China). The 16S rRNA genes of distinct regions (16S V4) were amplified using specific primer 515F (5′-GTGCCAGCMGCCGCGGTAA-3′) and 806R (5′-GGACTACHVGGGTWTCTAAT-3′) with the barcode. All PCR reactions were carried out with Phusion High-Fidelity PCR Master Mix (New England Biolabs, Ipswich, MA, USA). PCR products were mixed 1:1 (*v*:*v*) with loading buffer (containing SYBR green), followed by performing electrophoresis on 2% agarose gel. Samples illuminated with SYBR green between 400 and 450 bp were chosen for further experiments. The PCR products were purified with Qiagen Gel Extraction Kit (Qiagen, Hilden, Germany). Sequencing libraries were generated using TruSeq DNA PCR-Free Sample Preparation kit (Illumina, USA) following manufacturer’s recommendations with index codes. The quality of library was assessed by Qubit 2.0 Fluorometer and Agilent Bioanalyzer 2100 system (Thermo Scientific, Waltham, MA, USA). Then, the library was sequenced on an Illumina HiSeq 2500 platform and 250 bp paired-end reads were generated [50]. The original data have been uploaded to GEO database. The accession number for the high-throughput sequencing data reported is GEO: GSE139526.

### 4.4. Serum Collection and LPS Concentration Determination

Whole blood samples were collected from pregnant mice or E17 chick embryos. Serum was isolated by centrifugation (Thermo Scientific, Waltham, MA, USA) and stored at −80 °C. Serum LPS concentrations were measured by a chromogenic endpoint limulus amoebocyte lysate assay kit (Xiamen Horseshoe Crab Reagent Manufactory Co., Ltd., Xiamen, China).

### 4.5. Alcian Blue and Alizarin Red Staining

To visualize the skeleton, the whole chick embryos were stained with Alcian blue and alizarin red dyes as previously described. E17 chick embryos were freed from adherent tissues, fixed in 95% ethanol for 3 days and stained for cartilage with Alcian blue and counterstained for bone with alizarin red (Solarbio, Beijing, China). Long bone tissues were carefully photographed using a stereomicroscope (MVX10, Olympus, Tokyo, Japan). The length of each tibia, femur, humerus, ulna, radius, and phalange were quantified and analyzed using Image Pro-Plus 5.0 software (IPP, Media Cybernetics) [51].

### 4.6. Micro-Computed Tomography (Micro-CT)

The SkyScan 1172 system (Bruker micro-CT, Kontich, Belgium) was used for the CT scanning of chick phalanges as previously described with some modification [52]. For quantitative analysis, 1000 horizontal sections in total were selected to form OZ, namely selecting 500 horizontal sections on both sides of OZ, and trabecular bone regions were manually delineated adjacent to the trabecular bone outline. Image acquisition parameters used were: 5 µm voxel resolution, 0.5 mm aluminum filter, 80 kV voltage and 100 μA current, 750 ms exposure time, 0.4° rotation step, and using frame averaging. Reconstruction and analysis of images was performed using SkyScan software (Bruker micro-CT, Kontich, Belgium), NRecon (1.6.9.8), DataViewer (1.5.0), CTAnalyser (1.13.11.0) and CTvol (2.3.2.0).

### 4.7. Histology

The phalanges of the E17 chick embryos treated with saline or LPS were dismembered and fixed in 4% paraformaldehyde (PFA), then decalcified in a 10% EDTA solution in 1 mM PBS (pH 7.4) for 7 days at 4 °C, and finally embedded in paraffin. The samples were serially sectioned at 5 μm thickness on a microtome (Leica RM2126RT, Germany). Longitudinal sections of these bones were further stained with H&E (Sigma-Aldrich, St. Louis, MO, USA), Picrosirius red (Abcam, Cambridge, UK), von Kossa (Leagene, Beijing, China), and tartrate-resistant acid phosphatase (TRAP, Wako Pure Chemical Industries, Chuo-ku, Japan) following standardized protocols. The ratios of bone/type I collagen/mineralization area to total area of the slides were calculated and compared between two groups according to the methods previously reported with some modification [53]. That is, the staining areas of bone/type I collagen/mineralization were chosen according to their color on H&E, Picrosirius red, and von Kossa stained sections; total areas were selected by delineating the phalange contour, using IPP software.

### 4.8. Cell Culture

The osteogenetic cell line, MC3T3-E1 was obtained from ATCC (Manassa, VA, USA). The cells were cultured at 37 °C in a 5% CO_2_ incubator in αMEM supplemented with 10% FBS and 1% penicillin–streptomycin. Mineralization of MC3T3-E1 cultures was evaluated by micromass and monolayer cultures. Briefly, MC3T3-E1 cells were suspended in osteogenic differentiation medium at a density of 2.0 × 10^7^ cells/mL, spotted as 10 μL droplets per well on a six-well plate. After 3 h of pre-incubation, the cells were flooded with 500 μL of culture medium. The cells were incubated at 37 °C and 5% CO_2_ in an incubator (Galaxy S, RS Biotech, UK). LPS dissolved in osteogenic differentiation medium was introduced on the 2nd day after plating. The control cultures received PBS only. After treatment with LPS for 7 days, the cultures were fixed in 95% ethanol for 20 min and then were stained with 2% alizarin red dye (pH 4.2) at room temperature overnight to detect the calcium deposits (*n* = 3 wells for each group). For monolayer cell cultures, MC3T3-E1 cells were cultured in osteogenic differentiation medium at a density of 2.0 × 10^5^ cells per 35 mm dish. After treatment with LPS or PBS for 21 days, the cultures were stained with alizarin red dye as described above.

To examine the effect of LPS on osteoblastic differentiation, the MC3T3-E1 cells were induced by an osteogenic differentiation medium (αMEM with 10% FBS, 50 μg/mL ascorbic acid, and 5 mM β-glycerophosphate) for 7 days. These cells were also treated with 100 ng/mL LPS, 100 ng/mL LPS + 10^−6^ M AGN, 10^−6^ M RA, or PBS. The concentrations of LPS chosen were determined by the results shown in Appendix A.

### 4.9. Immunofluorescent Staining and F-Actin Staining

Immunofluorescent staining was performed on sections of the phalanges and MC3T3-E1 cells using polyclonal primary antibodies (Appendix A) at 4 °C overnight. These samples were then treated with Alexa Fluor 488 or 555 anti-rabbit or Alexa Fluor 488 anti-mouse IgG (1:1000, Invitrogen, Waltham, MA, USA) as a secondary antibody. For F-actin detection, the cultured cells were stained by phalloidin-Alexa-Fluor 488 or 555 (1:200, Invitrogen, Waltham, MA, USA) at room temperature for 2 h and then counterstained with 4′-6-diamidino-2-phenylindole (DAPI, 5 μg/mL; Invitrogen, Waltham, MA, USA) to reveal the nuclei, which were photographed using an epifluorescence microscope (Olympus IX51, Leica DM 4000B). The Image J software (National Institutes of Health, Bethesda, MD, USA) was used to quantitatively analyze the staining intensity of the protein expression.

### 4.10. RNA Isolation and qPCR

Total RNA was extracted from E17 chick phalanges and MC3T3-E1 cells using a Trizol kit (Invitrogen, Waltham, MA, USA). RNA (1 μg) was reverse transcribed, using 1 μL Oligo dT, 1 μL StarScript II RT mix, primer, and 10 μL 2× reaction mix, according to the manufacturer’s instructions (Genstar, Beijing, China). First-strand cDNA (0.4 μL) was synthesized to a final volume of 20 μL using a SuperScript RIII first-strand kit (Invitrogen, Waltham, MA, USA). Following reverse transcription, PCR amplification of the cDNA was performed using mouse/chick specific primers. The primers’ sequences are provided in Appendix A. The PCR reactions were performed in a Bio-Rad S1000TM Thermal cycler (Bio-Rad, Hercules, CA, USA) as described previously [54]. The expression of the genes was normalized to GAPDH, and the expression levels were compared by ΔΔCt. The qPCR results shown are representative of three independent experiments.

### 4.11. RNA Sequencing Transcript Profiling

The control and LPS-treated HH10 chick embryos were harvested, and mRNA was extracted then sequenced and analyzed by Shanghai Majorbio Bio-Pharm Technology Co., Ltd. (Shanghai, China). The original data have been uploaded to GEO database. The accession number for the RNA sequencing data reported is GEO: GSE139526.

### 4.12. Western Blotting

Proteins from chick phalanges and MC3T3-E1 cells were isolated using a radio-immuno-precipitation assay buffer (RIPA, Sigma–Aldrich, St. Louis, MO, USA). The concentration of protein was quantified with a BCA assay (Thermo Fisher Scientific, Waltham, MA, USA). Western blotting was performed in accordance with standardized procedures using specific antibodies (Appendix A). The loading control used was β-actin. Quantity One (Bio-Rad, Hercules, CA, USA) was used to capture the chemiluminescent signals and analyze the data. All samples were performed in triplicate.

### 4.13. Enzyme-Linked Immunosorbent Assay (ELISA)

ELISA kits (Meibiao Biol Tech, Jiangsu, China) were used to measure RA, TNF-α and IL-6 from the phalanges of the E17 chick embryos and MC3T3-E1 cells according to the manufacturer’s instructions. The results were calculated using interpolation from a standard curve created by a series of RA, TNF-α, and IL-6 concentrations.

### 4.14. Isolation and Purification of Matrix Vesicles (MVs)

MVs were released from MC3T3-E1 cell culture medium by enzymatic digestion with 2.5 mg/mL collagenase IA (Sigma, USA) and 2 mM CaCl_2_ for 2 h at 37 °C and 5% CO_2_. The digestion mix containing cells and MVs was centrifuged at 500× *g* for 5 min to pellet the cells. Supernatant was transferred to separate tube and centrifuged at 2000× *g* for 10 min to remove dead cells. Supernatant was again centrifuged at 10,000× *g* for 30 min to remove cell debris then at 100,000× g for 70 min to obtain the MV pellet. MV pellets were resuspended in PBS and centrifuged at 100,000× *g* for 60 min. Finally, the pellet was resuspended in 100 µL PBS and stored at −80 °C until further use [55].

### 4.15. Atomic Force Microscopy (AFM) Morphological Imaging and Membrane Ultrastructure Analysis

An atomic force microscope (BioScope Catalyst NanoScope-V, Veeco instruments, Santa Barbara, CA, USA) was combined with optical microscopy (IX 51, Olympus, Tokyo, Japan) to obtain topographical and ultrastructural changes of the MC3T3-E1 cells and MVs. All groups were imaged with ScanAsyst mode, which uses a patented curve acquisition method and complex algorithms for continuous monitoring of the quality of the image by adjusting the parameters automatically. The silicon nitride tips (150Al-G, Budget Sensors, Sofia, Bulgaria) were irradiated by ultraviolet for 15 min to remove any organic contaminants. The cells were exposed in air and images were captured at 256 × 256 pixels resolution and 1.0 Hz scan speed [56]. More than 10 cells or 100 MVs were measured in each group. All the topographical images were evaluated using Nanoscope analysis software 8.14. The cell surface arithmetic average roughness (Ra) and root-mean-square roughness (Rq) were calculated for different 5 × 5 µm^2^ areas of MC3T3-E1 cells.

### 4.16. Transmission Electron Microscopy (TEM)

Suspensions of MVs in 70% ethanol were placed on carbon-coated Cu grids and air-dried. The samples were analyzed using Joel 2100 TEM (Tecnai Spirit, FEI Company, Eindhoven, the Netherlands) at 200 kV bright field and selected area electron diffraction (SAED) modes. Images were acquired using a Gatan 832CCD camera (Gatan, Warrendale, PA, USA).

### 4.17. Data Analysis

Data analyses and construction of statistical charts were performed using the GraphPad Software (La Jolla, CA, USA). All data were expressed as the mean value ($\bar{x}$ ± SEM). Statistical evaluation was performed using analysis of variance or with a two-tailed *t*-test. Statistical significance was defined as *p* < 0.05. The data are indicated with (*) for *p* < 0.05, (**) for *p* < 0.01, and (***) for *p* < 0.001.

## Figures and Tables

**Figure 1 ijms-21-02518-f001:**
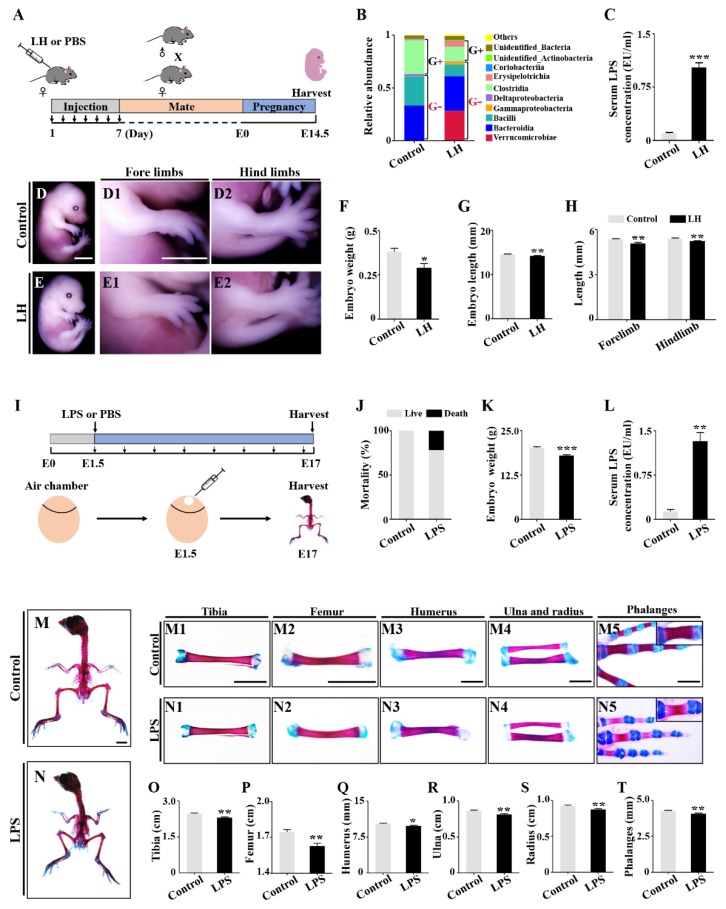
Assessing the development of mouse embryos from dysbacteriosis pregnant mice and the chick embryos treated with lipopolysaccharide (LPS). (**A**) Sketch illustrating the mouse model of lincomycin hydrochloride (LH)-induced dysbiosis of intestinal microbiota. (**B**,**C**) Bar charts showing the relative abundance of main bacteria in mouse guts at the class level of control and LH-treated groups (**B**) and the LPS level in mice serum (**C**). (**D**–**E2**) Representative bright-field images of E14.5 mouse embryos of control (**D**) and LH-treated (**E**) groups. High magnification images for forelimb (**D1**–**E1**) and hindlimb (**D2**–**E2**). (**F**–**H**) Bar charts showing embryo weights (**F**), lengths (**G**), as well as the lengths of forelimbs and hindlimbs (**H**) of control and LH-treated groups. (**I**) Sketch illustrating the model of LPS-treated chick embryos. (**J**–**L**) Bar charts showing embryo mortality (**J**), weights (**K**), as well as the serum LPS levels (**L**) of control and LPS-treated E17 chick embryos. (**M**,**N**,**M1**–**N5**) Representative Alcian blue/alizarin red whole-stained images of E17 chick embryos of control (**M**) and LPS-treated (**N**) groups. High magnification images of tibia (**M1**,**N1**), femur (**M2**,**N2**), humerus (**M3**,**N3**), ulna/radius (**M4**,**N4**), and phalanges (**M5**,**N5**). (**O**–**T**) Bar charts comparing the length of tibia, femur, humerus, ulna, radius, and phalanges between control and LPS-treated groups. Scale bars = 3 mm in (**D**–**E**,**D1**–**E1**,**D2**–**E2**,**M3**–**M5**,**N3**–**N5**); 10 mm in (**M**–**N**,**M1**–**M2**,**N1**–**N2**). (* *p* < 0.05, ** *p* < 0.01, *** *p* < 0.001).

**Figure 2 ijms-21-02518-f002:**
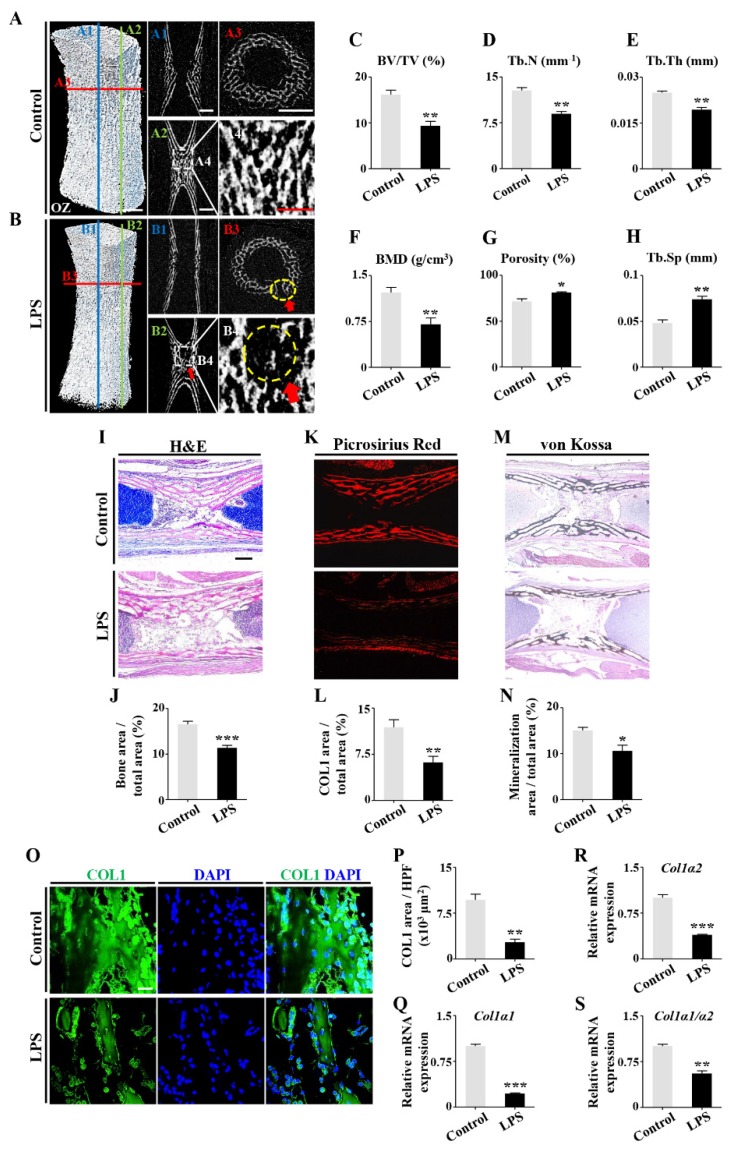
The assessment of osteogenesis in chick embryonic phalanges following LPS treatment. (**A**,**B**) The representative micro-CT images of the first phalanges of E17 chick embryos of control (**A**) and LPS-treated (**B**) groups. (**A1**–**A3**,**B1**–**B3**) are the longitudinal or transverse sections as indicated in (**A**,**B**), while (**A4**,**B4**) are high magnification images of the area indicated in (**A2**,**B2**). Yellow dotted circles in (**B3**,**B4**) show the rupture of and missing trabecular bone in the LPS-treated group. (**C**–**H**) Bar charts comparing the micro-CT parameters of bone volume fraction (BV/TV), trabecular numbers (Tb.N), trabecular thickness (Tb.Th), bone mineral density (BMD), porosity, and trabecular separation (Tb.Sp) between control and LPS-treated groups. (**I**–**M**) Representative images of H&E (**I**), Picrosirius red (**K**), and von Kossa (**M**) staining of control and LPS-treated groups. (**J**–**N**) Bar charts showing the ratios of area of bone (**J**), COL1 (**L**), and mineralization (**N**) of the total area between control and LPS-treated groups. (**O**) Representative immunofluorescent staining of COL1 and DAPI. Negative controls were shown in Appendix A (Appendix A). (**P**) Bar chart showing the area of COL1 per high-powered field (HPF) of control and LPS-treated groups. (**Q**,**S**) qPCR data showing the mRNA expression of *Col1α1* (**Q**), *Col1α2* (**R**), and the ratio of *Col1α1*/*Col1α2* (**S**) of control and LPS-treated groups. Scale bars = 0.5 mm in (**A**–**A3**,**B**–**B3**); 0.25 mm in (**A4**,**B4**, **I**–**M**); 20 μm in (**O**). OZ: ossification zone. (* *p* < 0.05, ** *p* < 0.01, *** *p* < 0.001).

**Figure 3 ijms-21-02518-f003:**
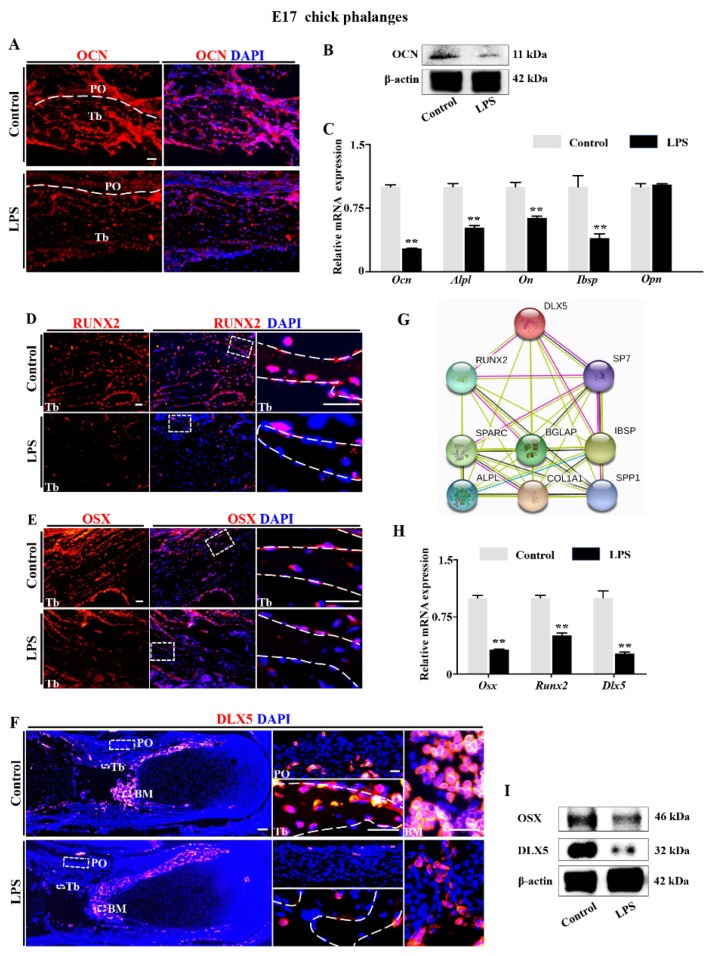
The assessment of the osteogenesis-related gene expressions following LPS exposure. (**A**) Representative images from immunofluorescent staining of OCN on the longitudinal sections of E17 chick phalange trabecular bone of control and LPS-treated groups, counterstained with DAPI. Dashed lines indicate the boundary between the periosteum (PO) and the trabecular bone (Tb). Negative controls were shown in Appendix A (Appendix A). (**B**,**C**) Western blot (**B**) and qPCR (**C**) showing the expression of OCN (**B**), as well as the expressions of *Alpl*, *On*, *Ibsp*, and *Opn* at mRNA level of control and LPS-treated groups. (**D**,**E**) The representative immunofluorescent staining of RUNX2 (**D**) and OSX (**E**) on the longitudinal sections of E17 chick phalange trabecular bone of control and LPS-treated groups. The areas outlined by dotted squares in trabecular bone are shown at a higher magnification in the right panels. The edge of trabecular bone is delineated by a dashed line. Negative controls were shown in Appendix A (Appendix A). (**F**) Representative immunofluorescent staining of DLX5 on the longitudinal sections of E17 chick phalanges of control and LPS-treated groups, counterstained with DAPI. Dotted squares in periosteum (PO), trabecular bone (Tb), and bone marrow (BM) are shown at higher magnification in right panels. The edge of trabecular bone is shown with a dashed line. Negative controls were shown in Appendix A (Appendix A). (**G**) The protein–protein interactions from the STRING database showing the network of DLX5, RUNX2, OSX (Sp7), OCN (BGLAP), IBSP, osteonectin (SPP1), COL1α1 (COL1A1), ALPL, and osteopontin (OPN or SPARC). (**H**) qPCR data showing the expressions of *Osx*, *Runx2*, and *Dlx5* of control and LPS-treated groups. (**I**) Western blot data showing the expressions of DLX5 and OSX of control and LPS-treated groups. Scale bars = 50 μm in (**A**,**D**,**E**,**F**). (** *p* < 0.01).

**Figure 4 ijms-21-02518-f004:**
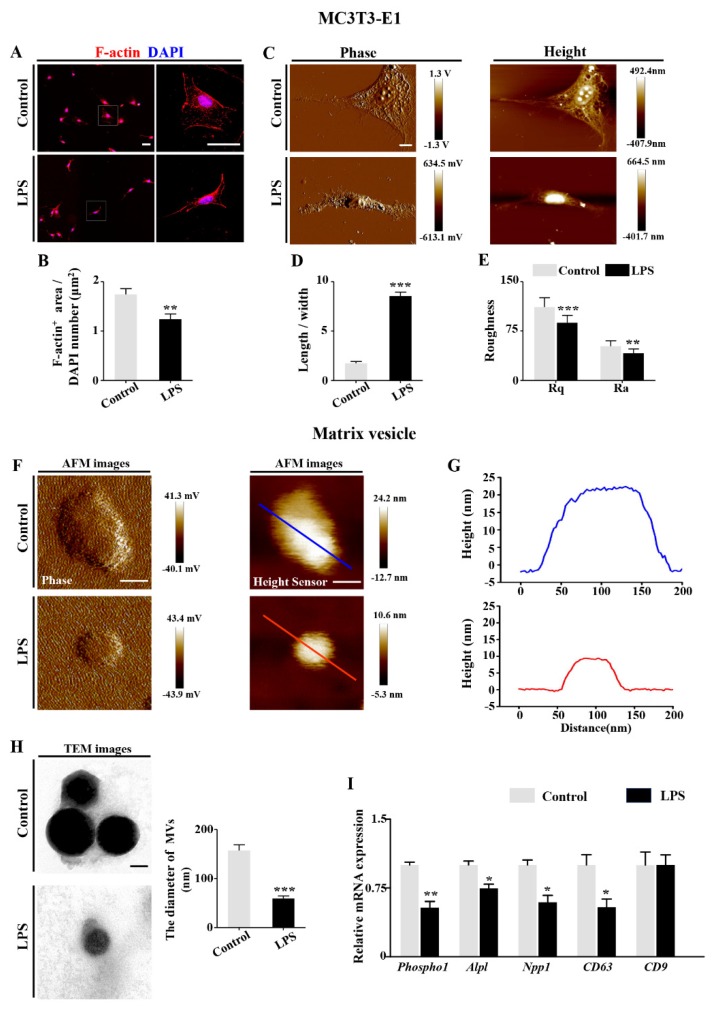
Assessment of the morphology of osteoblasts following LPS treatment. (**A**) Representative images of F-actin staining on 7-day cultured MC3T3-E1 cells of control and LPS-treated groups, counterstained with DAPI. The structures outlined by dotted squares are shown at a higher magnification in the right panels. Negative controls were shown in Appendix A (Appendix A). (**B**) Bar chart showing the ratio of F-actin-positive area/DAPI-positive cell numbers of control and LPS-treated groups. (**C**) Representative AFM phase and height images of MC3T3-E1 cells of control and LPS-treated groups cultured for 7 days. (**D**,**E**) Bar charts comparing the ratio of cell length to width (**D**), cell surface arithmetic average roughness (Ra) and root-mean-square roughness (Rq) (**E**) between control and LPS-treated groups. (**F**) Representative AFM phase and height sensor images of MVs collected from 7-day cultured MC3T3-E1 cells of control and LPS-treated groups. (**G**) Two-dimensional cross-sections of MVs taken across the maximum height of each vesicle. The lines in (**F**) show the direction of each cross-sectional measurement. (**H**) Representative TEM images of MVs secreted from MC3T3-E1 cells of control and LPS-treated groups. Bar charts at right panel show the comparison of diameter of MVs between control and LPS-treated groups. (**I**) qPCR data showing the expression of *Phospho1*, *Alpl*, *Npp1*, *CD63*, and *CD9* of control and LPS-treated groups. Scale bars = 50 μm in (**A**), 10 μm in (**C**); 50 nm in (**F** and **H**). (** *p* < 0.01, *** *p* < 0.001).

**Figure 5 ijms-21-02518-f005:**
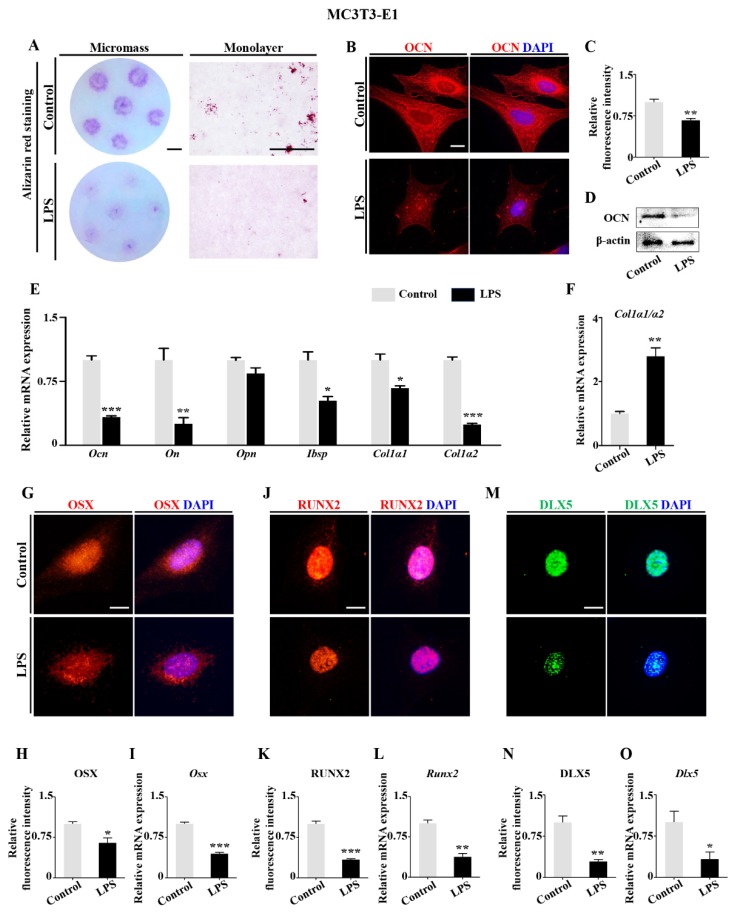
Assessment of the differentiation of MC3T3-E1 cells following LPS treatment. (**A**) Representative images of alizarin red staining of MC3T3-E1 cell mass cultures (left panels) or monolayer cultures after cultured for 21 days (right panels) of the control and LPS groups. (**B**,**C**) Representative images of OCN immunofluorescent staining on MC3T3-E1 cells counterstained with DAPI in control and LPS-treated groups Negative controls were shown in Appendix A (Appendix A). (**B**). OCN expression was quantified with fluorescence intensity (**C**). (**D**) Western blot data showing OCN expression of control and LPS-treated MC3T3-E1 cells. (**E**,**F**) qPCR data showing the expressions of *Ocn*, *On*, *Alpl*, *Ibsp*, *Col1α1*, *Col1α2*, and *Col1α1*/*Col1α2* in control and LPS-treated MC3T3-E1 cells. (**G**,**J**,**M**) Representative images of DLX5 (**G**), RUNX2 (**J**), and OSX (**M**) immunofluorescent staining on MC3T3-E1 cells, counterstained with DAPI. Negative controls were shown in Appendix A (Appendix A). (**H**,**K**,**N**) Fluorescent staining intensities of DLX5 (**H**), RUNX2 (**K**), and OSX (**N**) of control and LPS-treated MC3T3-E1 cells. (**I**,**L**,**O**) qPCR data showing the mRNA expression of *Dlx5*, *Runx2*, and *Osx* in control and LPS-treated MC3T3-E1 cells. Scale bars = 5 mm in (**A**); 20 μm in (**B**,**G**–**M**). (* *p* < 0.05, ** *p* < 0.01, *** *p* < 0.001).

**Figure 6 ijms-21-02518-f006:**
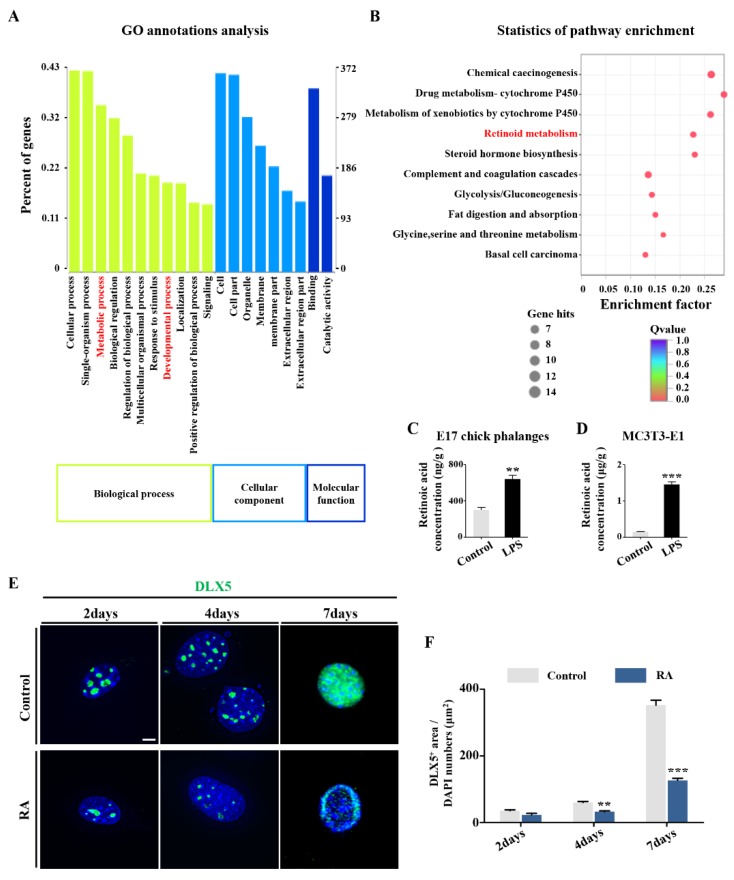
Determining the principal genes modulating osteogenesis following LPS treatment. (**A**,**B**) RNA sequencing transcript profiling showing the significantly altered signaling pathways identified using GO analysis (**A**) and KEGG analysis (**B**). (**C**,**D**) Bar charts showing RA concentration in E17 chick phalanges and MC3T3-E1 cells of control and LPS-treated groups. Units represent measured RA concentration normalized to total protein. (**E**) Representative images of DLX5 immunofluorescent staining of MC3T3-E1 cells from control and RA-treated groups, after being cultured for 2, 4, and 7 days, counterstained with DAPI. Negative controls were shown in Appendix A (Appendix A). (**F**) *Dlx5* expression in each cell was quantified with the ratio of DLX5-positive area to total DAPI-positive cell numbers. Scale bars = 6 μm in (**E**). (** *p* < 0.01, *** *p* < 0.001).

**Figure 7 ijms-21-02518-f007:**
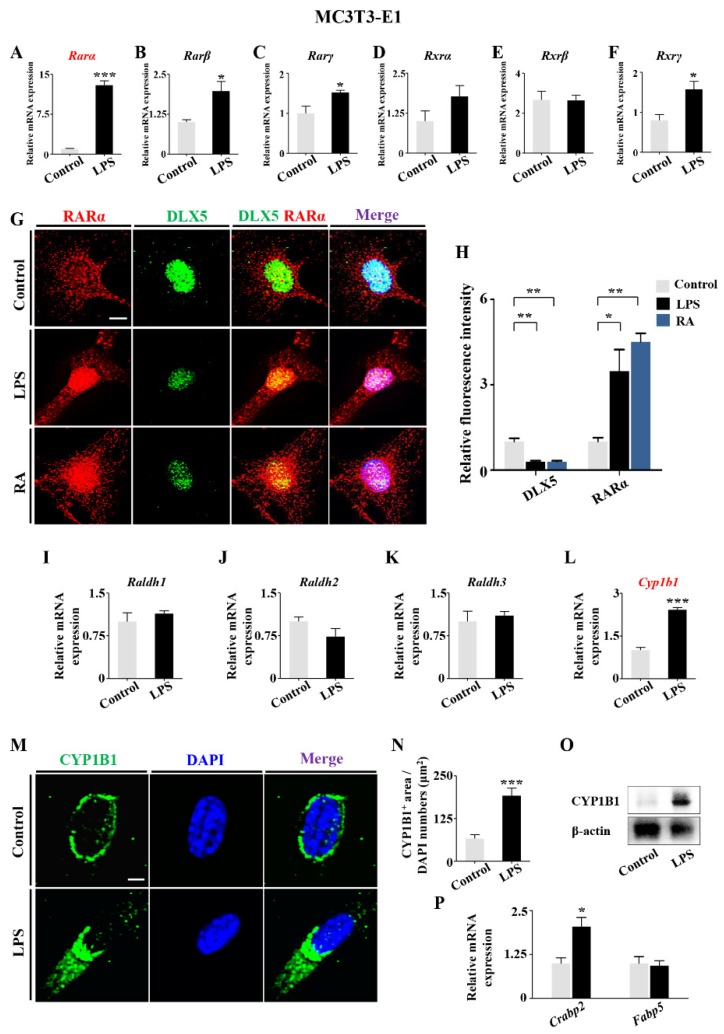
Determining the expressions of RA signaling and DLX5 following LPS treatment. (**A**–**F**) qPCR data showing the mRNA expression of *RARα* (**A**), *RARβ* (**B**), *RARγ* (**C**), *RXRα* (**D**), *RXRβ* (**E**), and *RXRγ* (**F**) in MC3T3-E1 cells of control and LPS-treated groups. (**G**) Representative images of RARα and DLX5 immunofluorescent staining in control, LPS-treated, and RA-treated MC3T3-E1 cells, counterstained with DAPI. Negative controls were shown in Appendix A (Appendix A). (**H**) Immunofluorescent intensity of RARα and DLX5 expressions was quantitatively measured in control, LPS-treated, and RA-treated MC3T3-E1 cells. (**I**–**L**) qPCR data showing the expression of *Raldh1* (**I**), *Raldh2* (**J**), *Raldh3* (**K**), and *Cyp1b1* (**L**) in MC3T3-E1 cells from control and LPS-treated groups. (**M**) Representative images of CYP1B1 immunofluorescent staining in control and LPS-treated MC3T3-E1 cells, counterstained with DAPI. Negative controls were shown in Appendix A (Appendix A). (**N**) Expression of CYP1B1 in each group was quantified with CYP1B1-positive area/total DAPI-positive cell numbers (**N**) in control and LPS-treated groups. (**O**) Western blot data showing the expression of CYP1B1 in MC3T3-E1 cells of control and LPS-treated groups. (**P**) qPCR data showing the mRNA expressions of *Crabp2* and *Fabp5* in control and LPS-treated MC3T3-E1 cells. Scale bars = 10 μm in (**G**); 6 μm in (**M**). (* *p* < 0.05, ** *p* < 0.01, *** *p* < 0.001).

**Figure 8 ijms-21-02518-f008:**
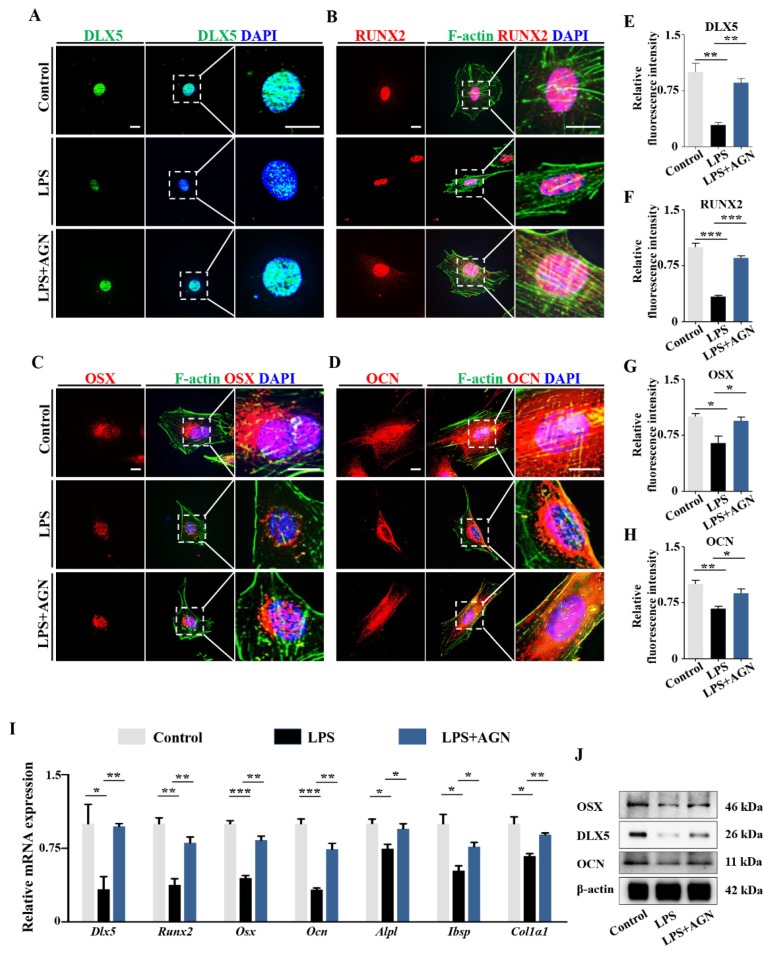
Determining the expression of osteogenesis-related genes following LPS treatment and blockage of RA signaling. (**A**–**D**) Representative images of DLX5 immunofluorescent staining on MC3T3-E1 cells, counterstained with DAPI (**A**), immunofluorescent staining of RUNX2 (**B**), Osx (**C**), and OCN (**D**), counterstained with F-actin and DAPI from control, LPS-treated, and LPS+AGN-treated groups. Negative controls were shown in Appendix A (Appendix A). (**E**–**H**) Fluorescent staining intensities of DLX5 (**E**), RUNX2 (**F**), OSX (**G**), and OCN (**H**) from control, LPS-treated, and LPS+AGN-treated groups. (**I**) qPCR data showing the mRNA expression of *Dlx5*, *Osx*, *Runx2*, *Ocn*, *Alpl*, *Ibsp*, and *Col1α1* in MC3T3-E1 cells from control, LPS-treated, and LPS+AGN-treated groups. (**J**) Western blot data showing the protein expressions of OSX, DLX5, and OCN in MC3T3-E1 cells from control, LPS-treated, and LPS+AGN-treated groups. Scale bars = 15 μm in (**A**–**D**). (* *p* < 0.05, ** *p* < 0.01, *** *p* < 0.001).

**Figure 9 ijms-21-02518-f009:**
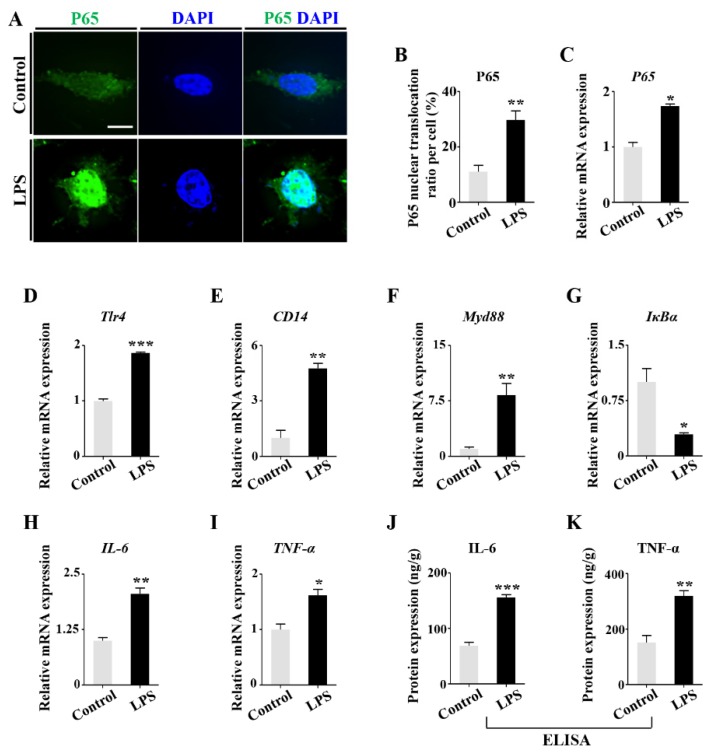
Determining the expression of the inflammatory factors following LPS treatment. (**A**) Representative images of P65 immunofluorescent staining on control and LPS-treated MC3T3-E1 cells, counterstained with DAPI. Negative controls were shown in Appendix A (Appendix A). (**B**) Bar chart comparing the P65 nuclear translocation ratios (percentage) in every cell between control and LPS-treated group. (**C**–**I**) qPCR showing the expressions of *P65*, *Tlr4*, *CD14*, *Myd88*, *IL-6*, and *TNF-α* genes at mRNA level of control and LPS-treated MC3T3-E1 cells. (**J**,**K**) ELISA assay showing the protein expression of IL-6 and TNF-α. Units represent measured protein normalized to total protein. Scale bars = 10 μm in (**A**). (* *p* < 0.05, ** *p* < 0.01, *** *p* < 0.001).

**Figure 10 ijms-21-02518-f010:**
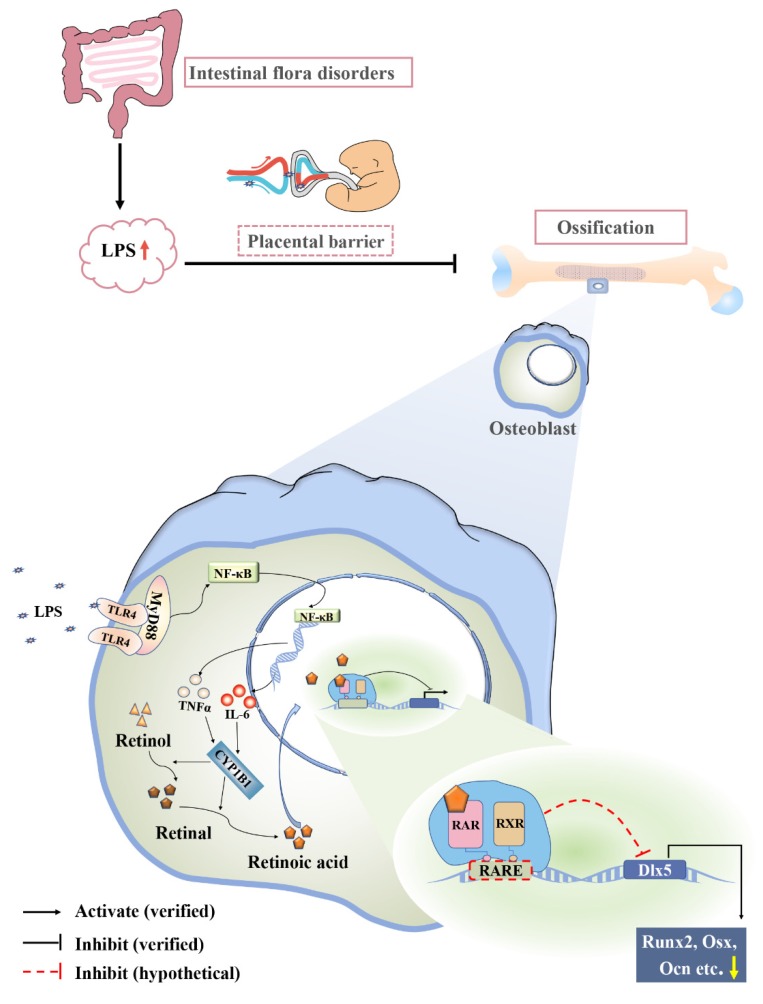
Proposed mechanism of RA-mediated *Dlx5* expression in abnormal osteogenesis induced by dysbacteriosis-induced LPS elevation.

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
