# Peer review of "Dysbacteriosis-Derived Lipopolysaccharide Causes Embryonic Osteopenia through Retinoic-Acid-Regulated DLX5 Expression"

_ijms, 2020, doi:10.3390/ijms21072518_

Round 1
Reviewer 1 Report
In the present work, the authors investigated the effect of dysbacteriosis on osteoblastogenesis and showed 1) that dysbacteriosis-derived LPS treatment leads to skeletal developmental retardation in vivo models. They observed 2) that LPS inhibited osteoblast differentiation. Then, through RNA-seq and pharmacological approaches, they found RA play a key role in LPS-induced osteopenia by downregulating Dlx5.
Comments to the authors:
Although the inhibition of osteoblast differentiation induced by LPS was already known by other papers, the authors found a detailed mechanism that LPS-induced RA inhibits osteoblastogenesis. The approaches to investigate the effect of dysbacteriosis-derived LPS on bone development are very straightforward. Overall, there are interesting discoveries and solid data in this work, most of which are cohesive and support the authors’ conclusions. Therefore, this reviewer supports its publication in IJMS.
Author Response
Point:
In the present work, the authors investigated the effect of dysbacteriosis on osteoblastogenesis and showed 1) that dysbacteriosis-derived LPS treatment leads to skeletal developmental retardation in vivo models. They observed 2) that LPS inhibited osteoblast differentiation. Then, through RNA-seq and pharmacological approaches, they found RA play a key role in LPS-induced osteopenia by downregulating Dlx5.
Comments to the authors
Although the inhibition of osteoblast differentiation induced by LPS was already known by other papers, the authors found a detailed mechanism that LPS-induced RA inhibits osteoblastogenesis. The approaches to investigate the effect of dysbacteriosis-derived LPS on bone development are very straightforward. Overall, there are interesting discoveries and solid data in this work, most of which are cohesive and support the authors’ conclusions. Therefore, this reviewer supports its publication in IJMS.
Reply: Thank you very much indeed for your comments.
Reviewer 2 Report
The authors present a well thought out study on the effect of microbiome disbacteriosis and specifically LPS during pregnancy on embryonic bone development with regards to osteoblast differentiation in a murine and chicken model.
Introduction:Well written and provides a good representation of the background knowledge. Here or in the discussion, the authors are advised to state why LPS injection in chicken embryos over injection in pregnant mice was used.
Results: Generally, very convincing. For the immunohistochemistries, the authors should provide negative staining control as small insert.
P5, Figure 2: Explain the area in the yellow, dotted circle in B3,4.
Discussion: Out of interest, have the authors also determined if the other effector cells of bone turnover: osteoclasts and osteocytes were impaired?
Methods: In general, very well explained, but please give more details where you state "as previously established".
P16, line 420: Please state clearly that saline injection was the control to treatment.
P17, line 463: Change to "von Kossa". Define Trap as abbreviation. And more details must be given to enable the reader to understand the detailed analysis. It is not enough to say, as previously published with some modification. Please expand.
P18, line 499: State the amount of mRNA reverse transcribed and the amount of cDNA used for qPCR.
P18, line 527: Give more details of MV preparation procedure.
P18, line 529: Change to "atomic force microscopy".
Author Response
Response to Reviewer 2 Comments
Point 1: The authors present a well thought out study on the effect of microbiome disbacteriosis and specifically LPS during pregnancy on embryonic bone development with regards to osteoblast differentiation in a murine and chicken model.
Response 1: Thank you very much indeed for the comments.
Point 2: Introduction: Well written and provides a good representation of the background knowledge. Here or in the discussion, the authors are advised to state why LPS injection in chicken embryos over injection in pregnant mice was used.
Response 2: Thank you very much indeed for the suggestion. Chick embryos can be cultured in vitro from stage HH0, which makes the manipulation of early stage embryos possible. Furthermore, the chick embryo works as a good model of skeletogenesis, since its skeletal size is larger than mice embryos of the same embryonic stage. These has been added into “Discussion” section, Page 14, Line 333-335.
Point 3: Results: Generally, very convincing. For the immunohistochemistries, the authors should provide negative staining control as small insert.
Response 3: Thank you very much indeed for your comment and suggestion.
Considering some of the pictures are quite small in current figures, so the negative staining control are organized together as supplementary figures (Figure S6 and S7). Figure S6 is negative control of tissue staining, while Figure S7 is negative control of cell cultures staining.
Point 4: P5, Figure 2: Explain the area in the yellow, dotted circle in B3,4.
Response 4: Thank you very much for your suggestion. The yellow dotted circle in B3-B4, are showing the rupture and missing of trabecular bone. This has been added into the legends of Figure 2 (Page 5, Line 142-143).
Point 5: Discussion: Out of interest, have the authors also determined if the other effector cells of bone turnover: osteoclasts and osteocytes were impaired?
Response 5: Thank you very much for raising the question. We checked the numbers and function of osteoclasts by TRAP staining and qPCR, when the phenotype of osteopenia was confirmed after the chick embryos were exposed to LPS. It was shown that osteoclasts were not negatively affected by LPS exposure significantly. The formation of bone is orchestrated by osteoblasts and osteoclasts. However, the role of osteoblasts surpasses that of osteoclasts during embryonic osteogenesis, the reason might partly attribute to without bearing body weight during embryonic stages, since we found that the peak of osteoclasts appeared after birth in chick embryos in our previous experiments.
As for osteocytes, actually we didn’t focus on that. That will be a good suggestion for future study definitely. Osteocytes function importantly during bone metabolism, which might be involved during LPS exposure, the analysis of RNA sequencing results also reveal the changes of several metabolic pathways. But we didn’t check osteocytes in current paper.
Point 6: Methods: In general, very well explained, but please give more details where you state "as previously established".
Response 6: Thank you very much for the suggestion. We have supplemented the details of the methods, especially those modified from publications, and the corresponding correction was added at Methods 4.3, 4.6, 4.7, 4.14, and 4.15 respectively. (Page 16, Line 445-456. Page 17, Line 473-476, Line 490-494. Page 19, Line 555-561, Line 565-570).
Point 7:P16, line 420: Please state clearly that saline injection was the control to treatment.
Response 7:Thank you very much for your suggestion. The information have been added into the revised manuscript, Page 16, Line 429-431.
Point 8:P17, line 463: Change to "von Kossa". Define Trap as abbreviation. And more details must be given to enable the reader to understand the detailed analysis. It is not enough to say, as previously published with some modification. Please expand.
Response 8:Thank you very much indeed for your suggestions.
- All the “Von kossa” in the revised manuscript have been replaced by “von Kossa”, including Figure 2M.
- The full name of “TRAP” – “tartrate-resistant acid phosphatase” has been added into main text at Page 17,Line 487.
- We have explained the details of the methods cited and modified from publications, and the corresponding correction was added at Methods 4.3, 4.6, 4.7, 4.14, and 4.15 respectively.
Point 9:P18, line 499: State the amount of mRNA reverse transcribed and the amount of cDNA used for qPCR.
Response 9:Thank you very much indeed for the suggestion. The amounts have been added in the revised manuscript, Page 17, Line 527-529.
Point 10:P18, line 527: Give more details of MV preparation procedure.
Response 10: Thank you very much indeed for your suggestion. The detailed description of MV preparation procedure has been added at Page 19, Line 555-561.
Point 11:P18, line 529: Change to "atomic force microscopy".
Response 11: Thank you very much indeed for your suggestion. All the “atom force microscopy” in the manuscript have been replaced by “atomic force microscopy”. The full name of AFM is given when it first appears in Main Text (Page 7, Line 192).